# Fusion Network for Change Detection of High-Resolution Panchromatic Imagery

**Wahyu Wiratama and Donggyu Sim *** 

Department of Computer Engineering, Kwangwoon University, Seoul 139701, Korea; wiratama@kw.ac.kr
* Correspondence: dgsim@kw.ac.kr; Tel.: +82-2941-6470

**Abstract:** This paper proposes a fusion network for detecting changes between two high-resolution panchromatic images. The proposed fusion network consists of front- and back-end neural network architectures to generate dual outputs for change detection. Two networks for change detection were applied to handle image- and high-level changes of information, respectively. The fusion network employs single-path and dual-path networks to accomplish low-level and high-level differential detection, respectively. Based on two dual outputs, a two-stage decision algorithm was proposed to efficiently yield the final change detection results. The dual outputs were incorporated into the two-stage decision by operating logical operations. The proposed algorithm was designed to incorporate not only dual network outputs but also neighboring information. In this paper, a new fused loss function was presented to estimate the errors and optimize the proposed network during the learning stage. Based on our experimental evaluation, the proposed method yields a better detection performance than conventional neural network algorithms, with an average area under the curve of 0.9709, percentage correct classification of 99%, and Kappa of 75 for many test datasets.

**Keywords:** change detection; convolutional network; deep learning; panchromatic; remote sensing

## 1. Introduction

Change detection is a challenging task in remote sensing, used to identify areas of change between two images acquired at different times for the same geographical area. Such detection has been widely used in both civilian and military fields, including agricultural monitoring, urban planning, environment monitoring, and reconnaissance. In general, change detection involves a preprocessing step, feature extraction, and classification or clustering algorithm to distinguish changed and unchanged pixels. To obtain a good performance, the selected classification or clustering algorithm plays an important role in the field of change detection.

In prior studies, statistical approaches have been proposed to identify a change [1–3]. A corresponding maximal invariant statistic is obtained by analyzing a suitable group of transformations leaving problem invariant [2]. Then, a general problem of testing equality among *M* covariance metrices in the complex-valued Gaussian case is analyzed for synthetic aperture radar (SAR) change detection. A sample coherence magnitude as a change metric has been proposed by [3]. A new maximum-likelihood temporal change estimation and complex reflectance change detection is used for SAR coherent temporal change detection. Currently, a classification or clustering is becoming one approach to be used for change detection in remote sensed images by employing supervised or unsupervised learning, respectively. Feature selection and feature extraction are important aspects in this approach. Several detection algorithms using two images have been proposed with different features for different types of applications [3–19]. The methods used for change detection have mostly been designed to extract changed features such as in a difference image (DI) [3–9], a local change vector [10], or a texture vector [11–13]. A DI is a common feature used

to represent a change in information through the subtraction of temporal images. Local change vectors have also been used by applying neighbor pixels to avoid a direct subtraction based on the log ratio. This method computes a mean value of the log ratio of temporal neighbor pixels. A texture vector [11–13] is employed to extract statistical characteristics. These changed features are then fed into a classification or clustering algorithm to determine changed/unchanged pixels. Some unsupervised change detection methods have been proposed based on the fuzzy c-mean (FCM) algorithm [14,16]. Such approaches are useful when labels in the training stages are unavailable. The learning algorithms in the aforementioned studies are based on observed data without any additional information, therefore, their application leads to overfitting for invariant changes. Furthermore, they do not yield a reasonably good performance in the change detection rates because they do not incorporate accurate information without supervision. Therefore, supervised change detection methods, such as a support vector machine (SVM) [11,16–18], have been proposed. The basic SVM can apply a binary classification to changed or unchanged pixels with texture information or using a change vector analysis. These algorithms are not perfect in terms of incorporating accurate and full statistical characteristics for large multi-dimensional data. Furthermore, they do not yield the best detection performance for new datasets.

Recently, a deep convolution neural network (DCNN) was developed to produce a hierarchy of feature-maps such as learned filters. The aforementioned DCNN can automatically learn a complex feature space from a huge amount of image data. A DCNN can achieve a superior performance compared to conventional classification algorithms. A restricted Boltzmann machine (RBM) [19], a convolutional neural network (CNN) [20–22], and deep belief networks (DBNs) [23] have been proposed for use in change detection. Such change detection algorithms based on deep learning yield a relatively good performance in terms of the detection accuracy. However, most can be categorized into front-end differential change detection using low-level features such as a difference image as a feature input of their networks, resulting in sensitivity to several deteriorated conditions caused by geometric/radiometric distortions, different viewing angles, and so on. This front-end differential change detection conducts an early feature extraction of two image inputs into a single-path network. In contrast, back-end differential detection methods by employing dual-path networks have been proposed for fusing higher-level features with a long short-term memory (LSTM) model [24] to avoid the use of low-level difference features such as a difference image. In addition, a Siamese convolutional network (SCN) [25–27] and dual-DCN (dDCN) [28] were also proposed to detect changed areas by measuring the similarity with high-level network features. These algorithms achieve a relatively good performance, although false negatives are still observed.

To reduce false positives and false negatives in change detection, a fusion network incorporating low- and high-level feature spaces in neural networks was proposed in this paper. For low-level differential features, the difference image is fed into the front-end differential DCN (FD-DCN). For a high-level differential feature, a back-end differential dDCN (BD-dDCN) is employed. In addition, a two-stage decision algorithm is incorporated for post-processing to enhance the detection rate during the inference stage. The intersection and union operations are employed to validate the change map. First, an intersection operation is used to avoid false positives. The second-stage decision operates a union by considering the local information of the first decision. This stage is developed to validate and repair the change map from the first decision. In addition, this study introduces a new loss function that combines a contrastive loss and weighted binary cross entropy loss function to optimize high- and low-level differential features, respectively. In our experiment, we found that the proposed algorithm can yield a better performance than existing algorithms by achieving an average area under the curve (AUC) of 0.9709, a percentage correct classification (PCC) of 99%, and a Kappa of 75 for several test datasets.

This work contributes three main key features as follows. (1) Unlike the mentioned existing works above, we propose a fusion network by combining a front- and back-end networks to perform the low- and high-level differential detection in one structure. (2) A combining loss function between

contrastive loss and binary cross entropy loss is proposed to accomplish fusion of the proposed networks in training stage. (3) The two-stage decision as a post-processing is presented to validate and ensure the changes prediction at the inference stage to obtain better the final change map.

This paper is organized into five sections. In Section 2, related studies are briefly described. Section 3 presents the proposed algorithm in detail. Section 4 describes and analyzes the experiment results. Finally, we provide some concluding remarks regarding this research.

## 2. Deep Convolutional Network and Related Studies on Change Detection

Deep neural architectures with hundreds of hidden layers have been developed to learn high-level feature spaces. The recently developed convolutional neural network (CNN) is a deep learning architecture that has been shown to be effective in image recognition and classification [29]. The CNN architecture employs multiple convolutional layers, followed by an activation function, resulting in the development of feature maps. The rectified linear unit (ReLU) is widely used as the activation function in many CNN architectures. To progressively gather global spatial information, the feature maps are sub-sampled by the pooling layer. The final feature maps are connected to a fully connected layer to produce the class probability outputs ($P_{class}$), as shown in Figure 1. During the training stage, an objective loss such as cross-entropy is computed. All of the weighting parameters of the network are updated to reduce the cost function using the back-propagation algorithm.

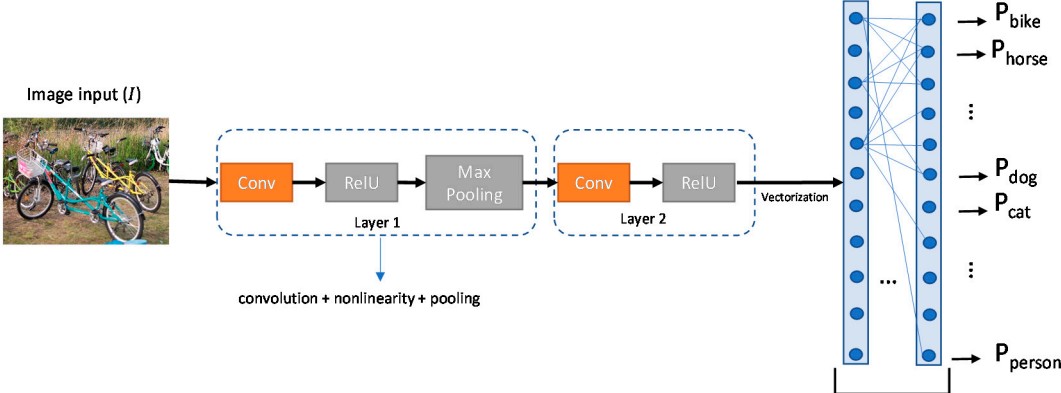

**Figure 1.** Convolutional neural network (CNN) architecture.

The related studies on change detection based on deep learning can be categorized into two categories based on the type of network that is used: A front-end differential network (FDN) and a back-end differential network (BDN). The front-end network uses low-level differential features such as a DI or joint feature (JF) as the feature input of the network, as shown in Figure 2a. In this case, a network with a single-path architecture receives the extracted DI as low-level differential features of the temporal images to identify changed pixels. Several studies based on an FDN have been proposed to improve the performance of the change detection rate. In addition, a deep neural network (DNN) is applied to detect objects from synthetic aperture radar (SAR) data [30]. The differential feature of temporal data is employed instead of a DI. This feature is used to solve the initial weight problem through pre-training using the restricted Boltzmann machine (RBM) algorithm. These pre-trained weights are then fed into the initial weights of the DNN during the training stage. In contrast, unsupervised change detection has been proposed by combining DBNs with a feature change analysis [23]. The feature maps of temporal input images are obtained using the DBN. The magnitude and direction of these feature maps are analyzed to distinguish the types of feature changes using an unsupervised fuzzy C-means algorithm. Other unsupervised systems have been proposed by combining a sparse autoencoder (SAE), unsupervised clustering, and a CNN to overcome the change detection problem without supervision [20]. First, a DI is computed using a log-ratio operator. The feature maps of the DI are extracted through the SAE and clustered into

change classes as the labels for the training CNN. Next, some feature maps extracted by the SAE are taken as the training data for the CNN. In addition, an autoencoder and multi-layer perceptron (MLP) are combined to identify changed pixels [31]. Change detection using faster R-CNN has been proposed for high-resolution images [32]. This work detects changed areas with bounding boxes. The DI is extracted and then fed into faster R-CNN to detect changed locations. Each of these deep learning algorithms tackles the change detection problem using a front-end differential network. This network identifies changes by observing low-level feature such as the DI, which is sensitive to various distortions, including geometric and radiometric distortions, and different viewing angles. Another approach of FDN to detect the changes has been proposed by joining feature inputs (JF) [23]. Two temporal images are concatenated and they are fed into DBN to avoid a DI for change detection. However, by joining the features in the early network causes both low-level differential inputs to be dependently learned in the single network. It is for global change detection, resulting in more false positives.

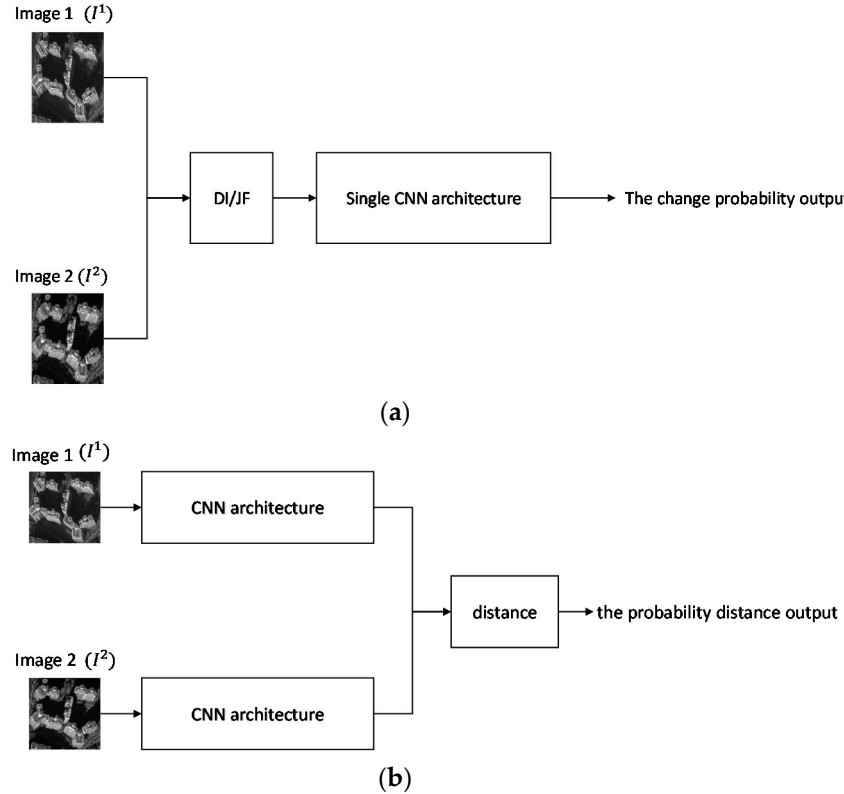

**Figure 2.** Front-end differential network (FDN) and back-end differential network (BDN) architectures: (**a**) Difference image (DI)/Joint features (JF) + single CNN, and (**b**) Dual-CNN.

Alternative algorithms for change detection were introduced by employing a high-level differential feature with a dual-path network, as shown in Figure 2b. Siamese CNN (SCNN) was proposed to detect changed areas for multimodal remote sensing data [27]. This architecture was employed to learn the different characteristics between multimodal remote sensing data. This approach learns the feature map of temporal images in each path network. The Euclidean distance was employed to measure the similarity at the back-end of the network. A similar method was developed based on an SCNN for optical aerial images [25]. A deep CNN was proposed by producing a change detection map directly from two images [33]. A change map was evaluated using the pixel-wise Euclidean distance from high-dimensional feature maps. Another method was proposed that incorporates a deep stacked denoising autoencoder (SDAE) and feature change analysis (FCA) for multi-spatial resolution change detection [34]. In the aforementioned study, denoising autoencoders were stacked to

learn local and high-level features for unsupervised learning. Then, the inner relationship between the multi-resolution image pair was exploited by building a mapping neural network to identify any change representations. A dual-dense convolutional network was presented by incorporating information from neighbor pixels [28]. In the aforementioned study, a dense connection was used to enhance the features of the changed map information. All of the above-mentioned BDN architectures yield good performances by inspecting high-level differential features, which can reduce false positives. However, a BDN can achieve higher sensitivity and specificity through high-level differential features.

Although a high-level differential network can improve the sensitivity and specificity, the false negative rate is still too high for practical applications. The FDN architecture can achieve a relatively higher true-positive rate regardless of the number of false positives. In addition, the BDN architecture can reduce the false-positive rate by producing some false negatives caused by strict decision criteria in high-level differential features. In this work, an FDN and a BDN were fused to employ the advantages of both. A post-processing step was then employed during the inference stage to obtain the final decision for change detection.

## 3. Proposed Fusion Network for Change Detection with Panchromatic Imagery

In general, a change detection system involves a pre-processing step to reduce geometric and radiometric distortions for better results. A radiometric correction is applied to remove atmospheric effects for a time-series image analysis. In this study, a radiometric correction was applied by converting digital numbers (DNs) into a radiance value. Then, the top-of-atmosphere (TOA) reflectance values were computed using the gain and offset values provided by a satellite provider. In addition, to ensure that the pixels in the image were in their proper geometric position on the Earth's surface, a geometric correction was applied. The parameters (polynomial coefficients) of the polynomial functions were estimated using least square fitting with ground control points (GCPs) identified in an unrectified image and corresponding to their real coordinates. A digital elevation model (DEM), namely, shuttle radar topography mission DEM (SRTM DEM), was then used to correct the optical distortion and terrain effect. The corrected images were then incorporated into the proposed network to detect changes.

To achieve a change detection, the proposed network employs a fusion network by fusing the FDN and BDN architectures. Dual outputs were generated to solve low-level differential and high-level differential problems. For the training stage, a contrastive loss function and a weighted binary loss function were combined to optimize the proposed fusion network parameters. In addition, a pre-processing step was applied to validate and ensure false changes during the inference stage. Intersection and union operations were then applied from the dual outputs of the proposed network. According to the proposed change detection, the false-positive and false-negative rates could be reduced, resulting in high sensitivity and specificity for a proper change detection. Symbols used in the proposed method are tabulated in Table 1.

**Table 1.** Symbols used in the proposed fusion network for change detection.

| Symbol | Description |
|---|---|
| $I^1$ and $I^2$ | Cropped temporal input image in time 1 and 2, respectively |
| $N^1$ and $N^2$ | Patch network 1 and 2 correspond to the back-end network |
| $N^3$ | Patch network 3 correspond to the front-end network |
| $F_{l,d_r}^i$ | Feature maps of the *l*-th layer at the *r*-th dense block and the *i*-th network |
| $P^1$ and $P^2$ | Outputs of $N^1$ and $N^2$, respectively |
| $D$ | Dissimilarity distance |
| $O$ | Change detection probability output of $N^3$ |
| $H_{l-1,d_r}^i$ | Incorporation process of a batch normalization (BN), a $3 \times 3$ convolution, and ReLU of the $(l-1)$-th layer at the *r*-th dense block and the *i*-th network |
| $[F_{0,d_r}^i,\ F_{1,d_r}^i,\ \ldots,\ F_{l-1,d_r}^i]$ | A concatenation of the feature-maps of all of previous layers, layer 0, . . . , and layer $(l-1)$ |
| $L$ | Proposed loss function |
| $E_c$ | Contrastive loss function |
| $E_B$ | Weighted binary cross entropy loss function |
| Y | Ground truth |
| $Ls$ and $L_D$ | Partial loss function for a pair of similar and dissimilar pixels, respectively |
| $m$ | Margin value |
| $\alpha$ | Weighted loss |
| $W$ | Proposed weighted function |
| $C$ and $U$ | Changed and unchanged numbers of pixels, respectively |
| $N$ | The number of full dataset |
| $\beta_c$ and $\beta_u$ | Penalization weights for false-negative and false-positive errors, respectively |
| $M_1$ | Change map for first prediction |
| $M_2$ | Change map for second prediction |
| $Nb$ | Local information of $M_1$ |
| T | Tested temporal images |
| $m$ and $n$ | Size of T |
| $s$ | Size of $I$ |

### 3.1. Fusion Network for Change Detection

For a change detection, an FDN architecture is commonly used for identifying changed pixels. Such an architecture uses low-level differential features that are relatively sensitive to noises. It is caused by direct low-level features comparison, which misalignments of geometric error and a different angle view are very influential. This FDN assigns a DI or JF to a single path network. They conduct dependent learning of both low-level features together which lead to hard learning for invariant changes and above-mentioned noisy conditions. Thus, this approach would produce a global change detection, resulting in true positives and more false positives. In addition, BDN architectures are designed to avoid low-level differential features, thereby reducing the false-positive detection rate. These architectures apply strict identification for a high-level differential, which may cause some false negatives. Therefore, an FDN is suitable in terms of the true-positive detection rate, and a BDN is extremely reliable for overcoming false positives. To obtain a proper change detection, a fusion network architecture is proposed by fusing an FD-DCN and a BD-dDCN with a dense-connectivity of the convolution layers, as shown in Figure 3. There are three branch networks, $N^1$, $N^2$, and $N^3$, receiving two temporal images ($I^1$ and $I^2$) in which $N^1$ and $N^2$ correspond to the back-end network, and $N^3$ refers to the front-end network by concatenating these two inputs ($I^1$ and $I^2$). A dense convolutional connection was employed in the proposed fusion network to enhance the feature representation [35]. This dense architecture is very effective at covering invariant change representations by reusing all preceding feature maps of the network. The proposed network was designed using dual outputs, namely, the dissimilarity distance ($D$) and change probability ($O$) at the last layer, corresponding to the back-end and front-end networks, respectively. Let us assume that the feature maps of the *l*-th layer at the *r*-th dense block and the *i*-th network are computed as:

$$F_{l,d_r}^i = H_{l-1,d_r}^i([F_{0,d_r}^i\ F_{1,d_r}^i,\ \ldots,\ F_{l-1,d_r}^i]),\ r = 0,1;\ i = 1,2,3, \tag{1}$$

where $[F_{0,d_r}^i,\ F_{1,d_r}^i,\ \dots,\ F_{l-1,d_r}^i]$ indicates a concatenation of the feature-maps of all of previous layers, layer 0, ... , and layer $(l-1)$. In addition, $H(\cdot)$ incorporates a batch normalization (BN), a $3 \times 3$ convolution, and ReLU. A pair of temporal images were cropped into two patches ($40 \times 40$) ($I^1$ and $I^2$) by sliding the window and fed into $N^1$ and $N^2$, respectively. The dissimilarity distance (D) was then computed based on the Euclidean distance, which is defined as follows:

$$D = \|P^1 - P^2\|_2 \tag{2}$$

where $P^1$ and $P^2$ are the outputs of $N^1$ and $N^2$ activated by sigmoid function, respectively. The proposed method applies a pixel-wise change detection by inspecting the neighboring pixels. The $40 \times 40$ patch images identify a change corresponding to the center pixel of the patch. Thus, when the value of $D$ is close to 1, the center of $I$ is assigned to a changed pixel. In addition, $I^1$ and $I^2$ were concatenated to be fed into $N^3$. The same dense convolution architecture was employed in this branch network to generate the change detection probability ($O$). The dual outputs ($D$ and $O$) are a result of this fusion network. In addition, a post-processing step during the inference stage was proposed based on these outputs ($D$ and $O$) to achieve a proper prediction.

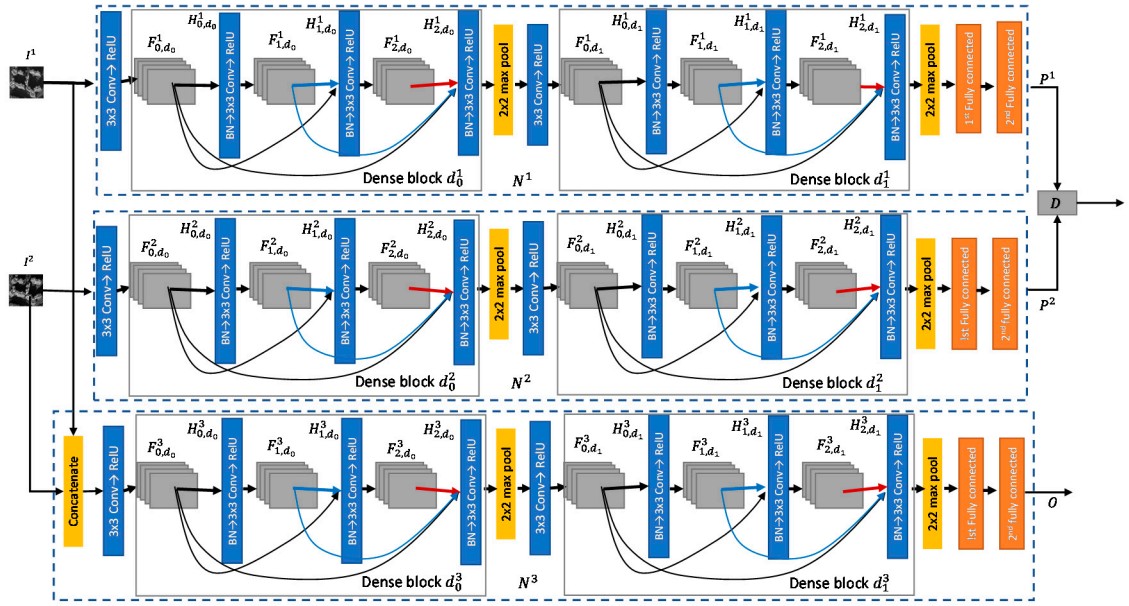

**Figure 3.** The proposed fusion network architecture for change detection.

### 3.2. Training of the Proposed Fusion Network for Change Detection

During the training stage, this paper introduced a loss function ($L$) by combining the contrastive loss ($E_c$) [36] and weighted binary cross entropy loss ($E_B$) as defined by:

$$L = \alpha E_c + (1 - \alpha)E_B \tag{3}$$

where $\alpha$ is a weight loss. Given a training set consisting of $40 \times 40$ image pairs and a binary label of the ground truth ($Y$), the proposed network was trained using backpropagation. Here, $E_c$ was applied to optimize the parameters of $N^1$ and $N^2$, and is as computed as follows [36]:

$$E_c = \sum_i (1 - y_i)L_S(D_i) + (y_i)L_D(D_i) \tag{4}$$

where $y = 1$ is a changed pixel and $y = 0$ is an unchanged pixel. In addition, $Ls$ is a partial loss function for a pair of similar pixels, and $L_D$ is a partial loss function for a pair of dissimilar pixels, as defined by [36]:

$$Ls = \frac{1}{2}(D_i)^2 \tag{5}$$

$$L_D = \frac{1}{2}(max\{0, m - D_i\})^2 \tag{6}$$

The value of $m$ is set to 1 as the margin value. In addition, $E_B$ was used to optimize the parameters of $N^3$, as defined by:

$$E_B = - \sum_i W_i(y_i \log(O_i) + (1 - y_i) \log(1 - O_i)) \tag{7}$$

where $W$ is the proposed weighted function used to penalize the false-positive and false-negative errors. Thus, $W$ is computed by:

$$W_i = y_i \left( \beta_c \left( 1 - \frac{C}{N} \right) \right) + (1 - y_i) \left( \beta_u \left( 1 - \frac{U}{N} \right) \right) \tag{8}$$

where $\beta_c$ and $\beta_u$ are penalization weights for false-negative and false-positive errors, respectively. Moreover, $C$ and $U$ are the changed and unchanged numbers of pixels in the full dataset ($N$), respectively.

The proposed network was trained using a stochastic gradient descent (SGD) with the training parameters, including 0.001, $1 \times 10^{-6}$, and 0.9 as the learning rate, decay, and momentum, respectively. In addition, the epoch number was set to 30. The value of $\alpha$ was set to 0.7 to further penalize $E_c$. It was to prevent false positives, which are possible in a back-end network. The goal of prediction through the front-end was to obtain better true-positive rates regardless of the number of false positives. Thus, the false negatives were penalized ten times more than false positives, namely, $\beta_c = 10$ and $\beta_u = 1$.

### 3.3. Dual-Prediction Post-Processing for Change Detection

During the inference stage, post-processing was introduced using dual-prediction for change detection. In the counting rule, binary hypotheses can be passed to a fusion center, which then decides which one of the two hypotheses is true [37]. The proposed algorithm employed a hard-logical rule using an AND and OR operation with the same probability output thresholds to predict a changed pixel. This aimed to validate and ensure the change detection based on the proposed fusion network outputs ($D$ and $O$). There were two steps to applying this post-processing. First, an intersection operation was employed to obtain a strict prediction and avoid false positives. Assume that ($m \times n$) images (T) will be tested using the proposed fusion network, resulting in an ($m \times n$) change map ($M_1$). This prediction was conducted by sliding in the raster scan order, as shown in Figure 4. The inputs ($I^1$ and $I^2$) with the central pixel position, x and y, were assigned to the proposed fusion network to generate the values of $D$ and $O$. If $D$ and $O$ identified a changed pixel, then $M_1(x, y)$ was set to a value of 1; otherwise, it was set 0. This was performed for the entire image T.

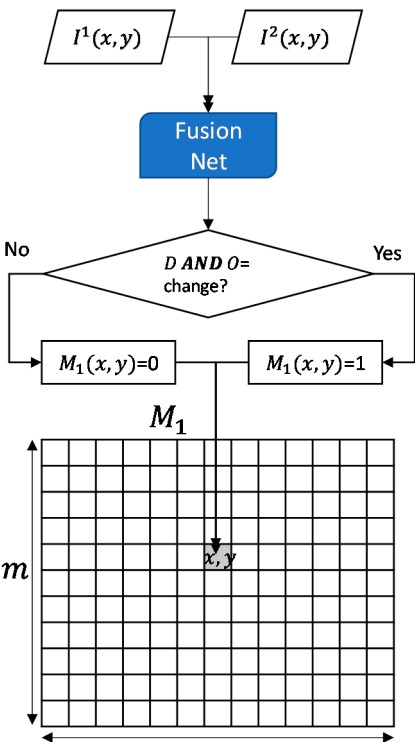

**Figure 4.** First prediction flowchart.

Then, the second prediction was performed to ensure the first prediction, as shown in Figure 5. Let us assume that $(m \times n)$ $M_2$ was a change map for the second prediction. Initially, a prediction noise was investigated by analyzing the local information from $M_1$ by computing $Nb$, as defined by:

$$Nb(x,y) = \sum_{i=x-\frac{q}{2}}^{x+\frac{q}{2}} \sum_{j=y-\frac{q}{2}}^{y+\frac{q}{2}} M_1(i,j). \tag{9}$$

where $Nb(x,y)$ computes the local information $M_1(x,y)$ using a $q \times q$ window. If the value of $Nb(x,y)$ is greater than the input size $s$ (40) divided by 4, then the second prediction is applied, otherwise, $M_2(x,y)$ is assigned to 0. A union operation was operated from $D$ and $O$ for the second prediction. When it returned the changed pixel, $M_2(x,y)$ was assigned a value of 1, otherwise, it was assigned a value of 0. The final change map was obtained based on the result of $M_2$.

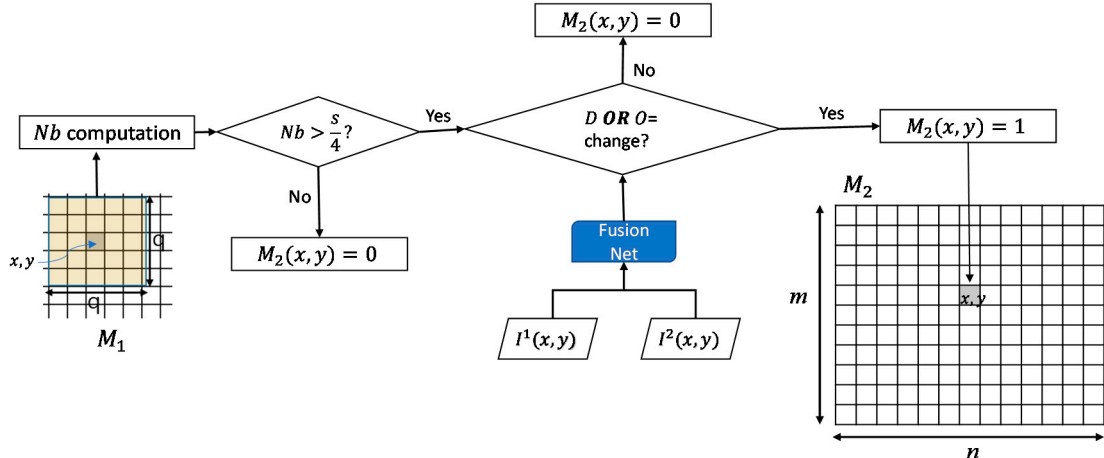

**Figure 5.** Second prediction flowchart.

## 4. Experimental Evaluation and Discussion

This study used a dataset of panchromatic imageries, which provided 0.7 GSD captured by the KOMPSAT-3 sensor. For the training dataset, this study used a scene of overlapped images (1214 × 886) over Seoul, South Korea, as shown in Figure 6. These images were cropped into a 40 × 40 sliding patch, and the center pixels of the cropped patch pair were labeled based on the ground truth.

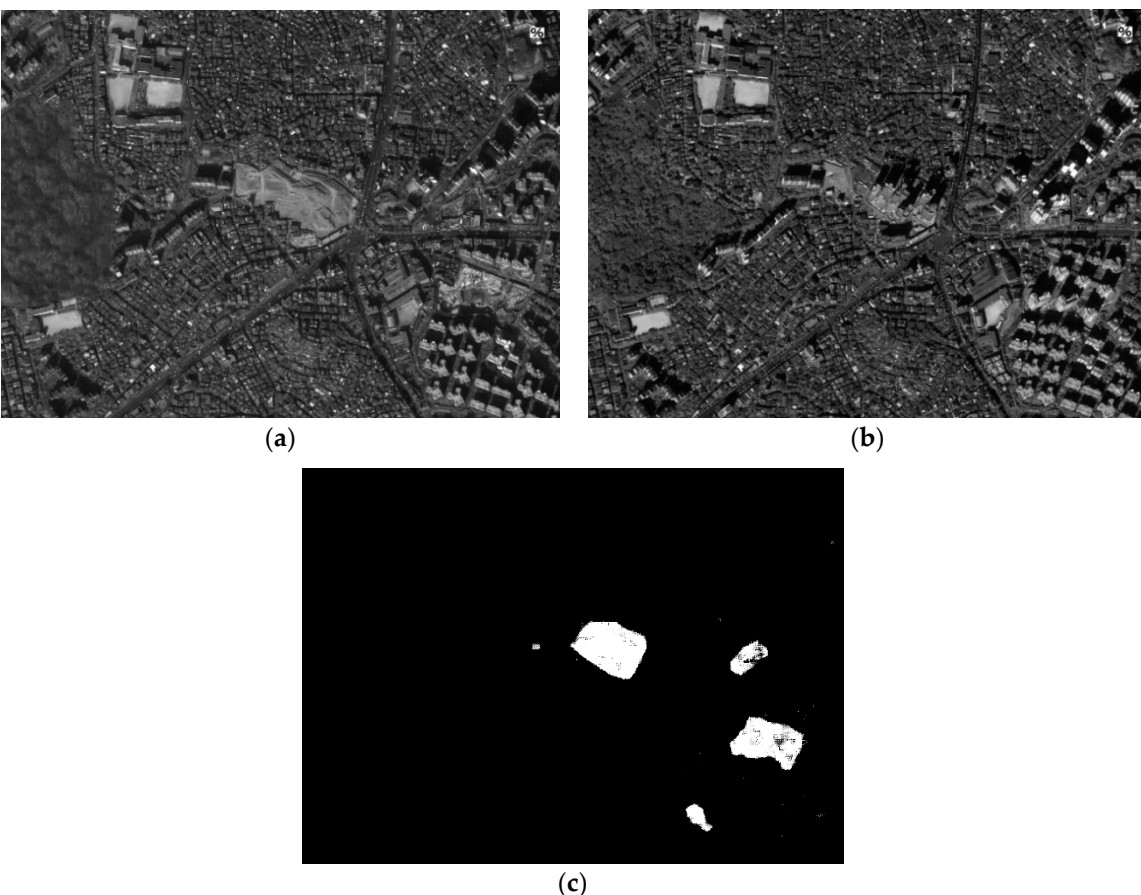

**Figure 6.** Training dataset: (**a**) Image acquired in March 2014, (**b**) image acquired in December 2015, and (**c**) the ground truth.

Figure 6 shows an area containing completed changes and changes under contraction. In addition, these images have many tall buildings, roads, houses, and forests to be trained for solving the misalignment and viewing angle problems. In our experiments, to assess the effectiveness of the proposed change detection system, three areas of the panchromatic datasets were used, namely, Areas 1, 2, and 3, as shown in Figure 7.

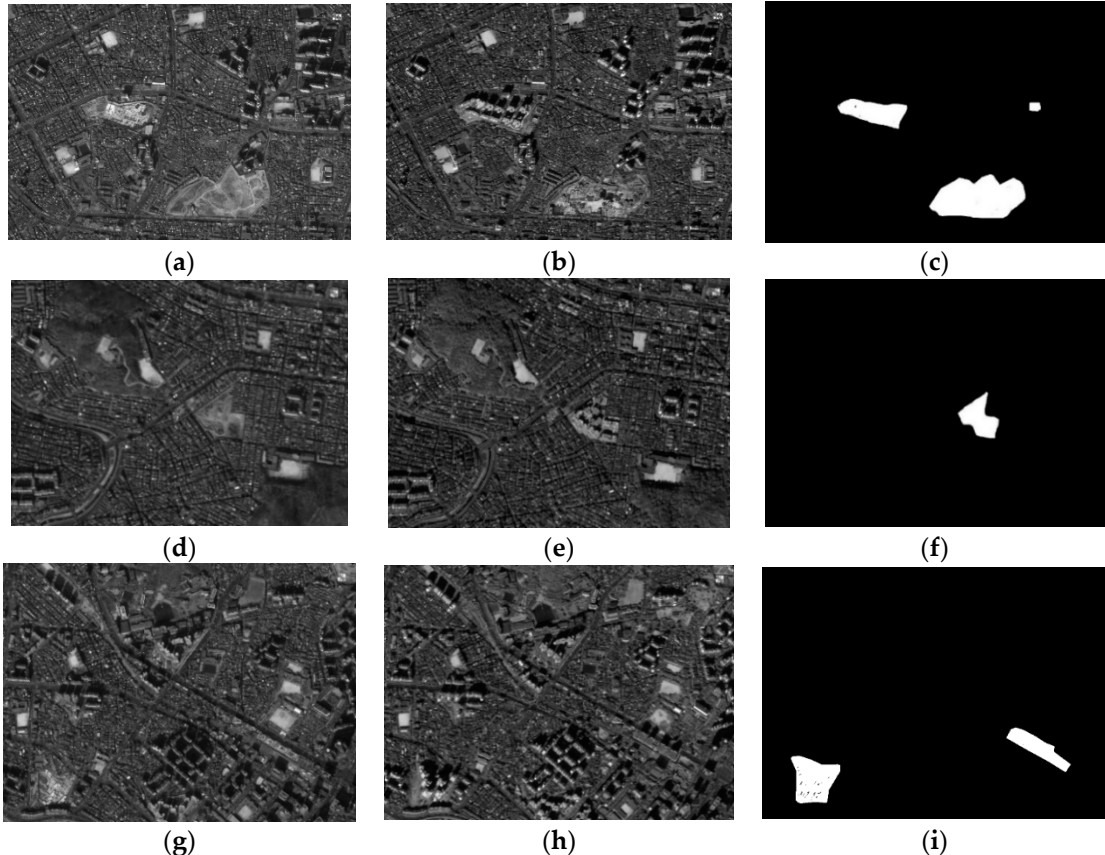

**Figure 7.** Experiment dataset: (**a**) Input image for Area 1 (March 2014), (**b**), input image for Area 1 (October 2015), (**c**) ground truth for Area 1, (**d**) input image for Area 2 (March 2014), (**e**) input image for Area 2 (October 2015), (**f**) ground truth for Area 2, (**g**) input image for Area 3, (March 2014), (**h**) image input for Area 3 (October 2015), and (**i**) ground truth for Area 3.

The images in Figure 7 were acquired in March 2014 and October 2015 over different areas of Seoul, South Korea. Each image pair had been radiometrically corrected and had a geometric misalignment of approximately ±6 pixels. In addition, it also had a different angle view, which cannot be resolved without precise 3D building models. Area 1 was located in a downtown part of Seoul, and contained areas changed through building construction. Moreover, the urban area had tall buildings and roads. These datasets included several factors of geometric distortion, misalignments, and different viewing angle effects, which could lead to many false changes. In addition, Area 2 represented a downtown area near a forest. These two images were acquired in different seasons. It was difficult to achieve robustness to seasonal changes for practical applications. Area 3 had many tall buildings, making it difficult to achieve an accurate detection rate owing to the different viewing angles.

In this study, the receiver operating characteristic (ROC) curve, AUC, PCC, and Kappa coefficient were used to quantitatively evaluate the performance of the proposed method. Moreover, to evaluate the effectiveness of the proposed method, it was compared with conventional algorithms having FDN and BD-dDCN architectures [28]. A DI and JF were incorporated into a single-path CNN architecture (DI + CNN and JF + CNN). These architectures included eight depth convolutional, two pooling, and two fully connected layers, which were the same as the proposed depth layers. In addition, Dual-DCN [28] was also compared to the proposed method.

Figure 8 shows an ROC curve, which indicates that the proposed method could achieve a better AUC compared to the existing algorithms. For Area 1, the proposed method yielded an AUC of 0.9904, which means that it could identify changes approximating the ground truth. It had a slightly higher dual-DCN of 0.9878. The FDN architectures provided an AUC lower than the proposed algorithm

which JF + CNN and DI + CNN gave an AUC of approximately 0.9509 and 0.7060, respectively. Furthermore, the proposed method significantly outperformed the other algorithms with regard to the AUC for Areas 2 and 3 because it could properly detect the change events with the incorporation of low- and high-level differential features. Table 2 summarizes the PCC and Kappa values of the different methods applied for the three areas. The proposed method showed a higher PCC in Areas 1 and 3. The dual-DCN achieved a slightly higher PCC than the proposed method in Area 2. However, in terms of the Kappa value, the proposed fusion network outperformed all other existing algorithms. The proposed method achieved a Kappa value of 75.16 on average, which means that it yielded a good agreement in terms of the results.

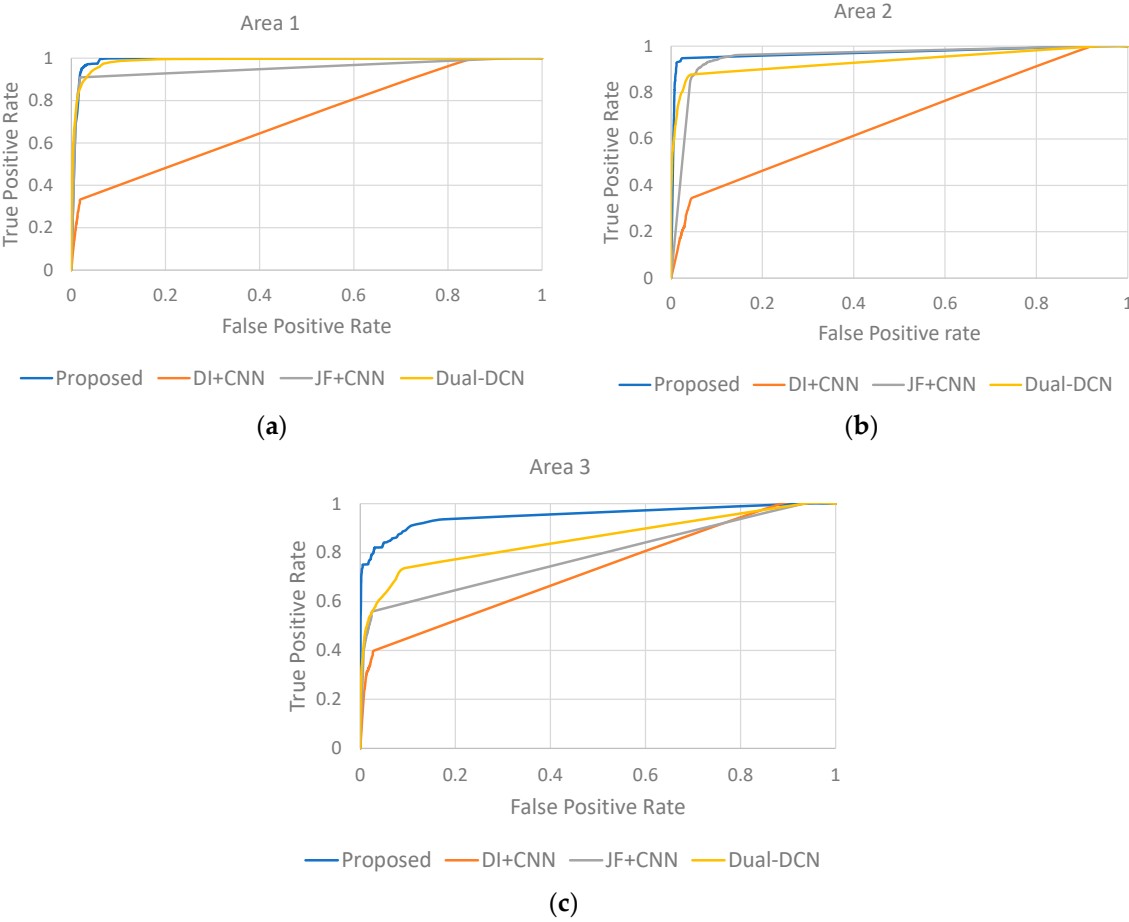

**Figure 8.** Receiver operating characteristic (ROC) for (**a**) Area 1, (**b**) Area 2, and (**c**) Area 3.

**Table 2.** Quantitative assessment of the existing and proposed algorithms.

| Algorithm | Area 1 | | | Area 2 | | | Area 3 | | |
|---|---|---|---|---|---|---|---|---|---|
| | **AUC** | **PCC** | **Kappa** | **AUC** | **PCC** | **Kappa** | **AUC** | **PCC** | **Kappa** |
| DI + CNN | 0.7060 | 0.9458 | 36.8938 | 0.6764 | 0.9571 | 11.8939 | 0.7213 | 0.9855 | 33.2651 |
| JF + CNN | 0.9509 | 0.9775 | 79.7190 | 0.9536 | 0.9570 | 29.7251 | 0.7847 | 0.9732 | 47.6066 |
| Dual-DCN | 0.9878 | 0.9774 | 78.4277 | 0.9546 | 0.9922 | 60.0070 | 0.8515 | 0.9751 | 50.7542 |
| Proposed | 0.9904 | 0.9782 | 80.7942 | 0.9707 | 0.9902 | 65.9929 | 0.9517 | 0.9892 | 78.6898 |

Figure 9 shows the change map results when applying the existing and proposed algorithms. Visually, the proposed method achieved a much better result than the existing algorithms. In Area 1, the proposed fusion network nearly approximated the ground truth. It could reduce the number of false positives while preserving the true positives. The proposed network produced a cleaner change map than the existing algorithms regarding false positives. Moreover, the proposed algorithm yielded reasonably good results for Areas 2 and 3. The proposed method significantly reduced the

number of false positives and enhanced the true positives. This is caused by the proposed fusion network, which was designed and trained for low- and high-level differential problems. In addition, a post-processing step was employed to validate and repair the change map.

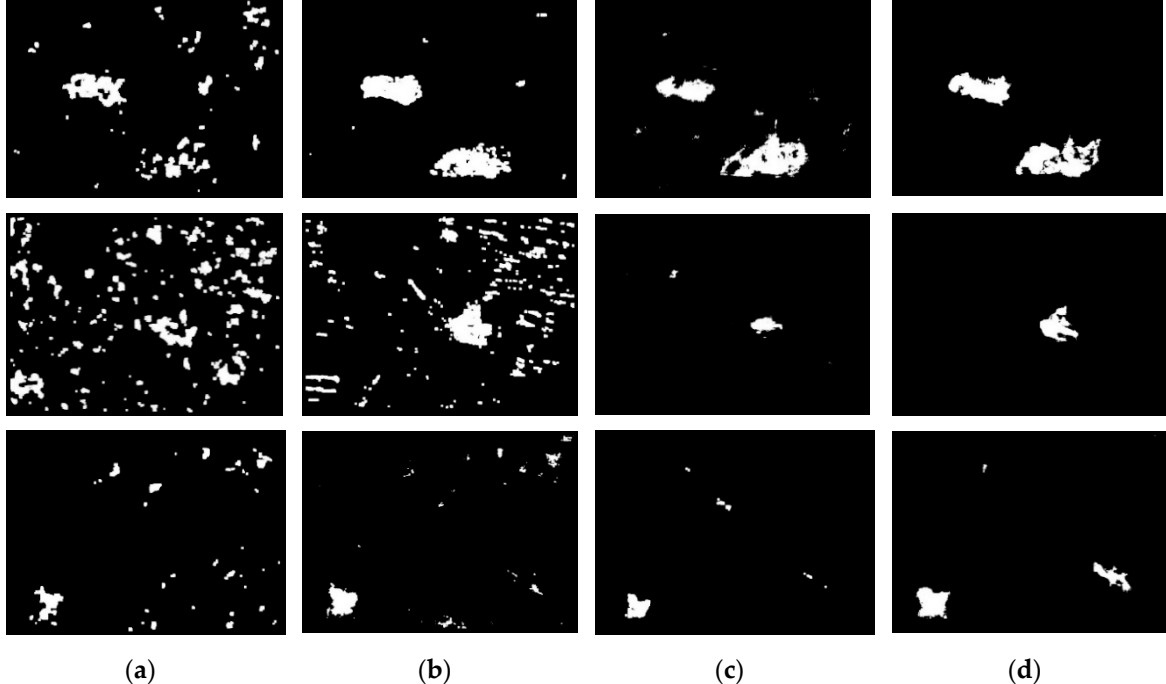

**Figure 9.** Detection results for three areas when using the existing and proposed algorithms: (**a**) DI + CNN, (**b**) JF + CNN, (**c**) dual-DCN, and (**d**) the proposed fusion network.

To evaluate the effectiveness of the proposed two-stage decision, the proposed algorithm also was compared to each individual network output (*D* and *O*) and the other decision method between two outputs of the proposed fusion network based on the mean operation. In addition, another single output fusion network (SOFN) architecture was designed same as the proposed fusion network architecture by fusing *D* and *O* outputs for more comparisons. This network was trained with the binary cross entropy loss function by the same training parameters. The objective and subjective evaluation are presented in Table 3 and Figure 10, respectively.

**Table 3.** Quantitative assessment of single output decision and proposed algorithms.

| Network Outputs | Area 1 | | | Area 2 | | | Area 3 | | |
|---|---|---|---|---|---|---|---|---|---|
| | AUC | PCC | Kappa | AUC | PCC | Kappa | AUC | PCC | Kappa |
| D | 0.9206 | 0.9655 | 65.3497 | 0.8154 | 0.9854 | 33.8273 | 0.8410 | 0.9794 | 55.1115 |
| O | 0.9357 | 0.9607 | 61.9808 | 0.8948 | 0.9879 | 50.0476 | 0.8667 | 0.9436 | 31.3879 |
| Mean | 0.9886 | 0.9781 | 78.4481 | 0.9588 | 0.9875 | 52.7712 | 0.9165 | 0.9803 | 59.5712 |
| SOFN | 0.9685 | 0.9661 | 65.7595 | 0.8903 | 0.9897 | 52.6660 | 0.8658 | 0.9820 | 61.2094 |
| Proposed | 0.9904 | 0.9782 | 80.7942 | 0.9707 | 0.9902 | 65.9929 | 0.9517 | 0.9892 | 78.6898 |

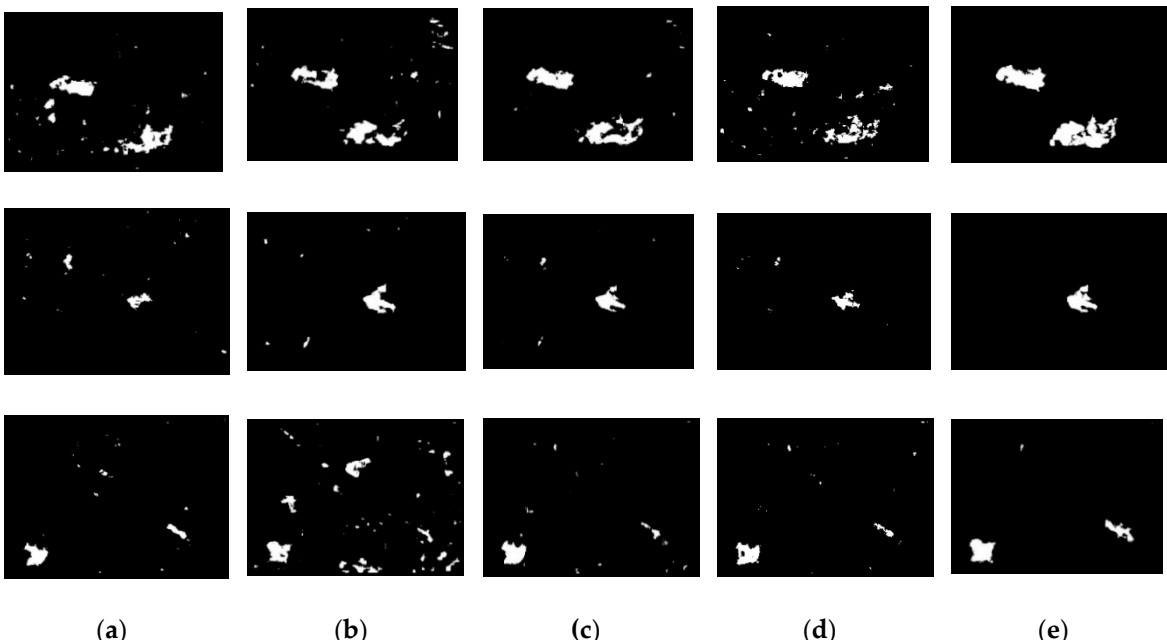

<center>(<b>a</b>)　　　　　　(<b>b</b>)　　　　　　(<b>c</b>)　　　　　　(<b>d</b>)　　　　　　(<b>e</b>)</center>

**Figure 10.** Detection results for three areas when using an individual network output and the proposed algorithms: (**a**) *D*, (**b**) *O*, (**c**) Mean, (**d**) SOFN, and (**e**) the proposed fusion network with a two-stage decision.

According to Table 3, the proposed two-stage decision shows better performance compared to individual outputs and mean operation. In term of AUC, PCC, and Kappa, the proposed gave significantly better results than that by individual outputs (*D* and *O*). Figure 10 shows that the output *O* produced more true positives regardless of the number of false positives. However, the output *D* can reduce the false-positive rate. This condition makes the proposed two-stage decision working as the goal that detection rates can be accelerated by the combining of two network outputs with a two-stage decision. In addition, the proposed algorithm still outperformed the mean operation between two network outputs for all areas. SOFN with the single output also gave worse results than the proposed one caused by no validation decision of post-processing for change detection. The proposed fusion network was employed with a two-stage decision to obtain a better prediction rate.

Regarding time complexity, the proposed fusion network consumed more computational complexity than the existing algorithm by a factor of approximately two over the dual-path network and three with the single-path network. It was due to the proposed architecture designed with more network paths. In addition, the proposed two-stage decision required an additional prediction process in the inference stage. Let us see that the general total time complexity of dense convolutional network [35] was $O(K^2)$ run-time complexity for a depth *K* network [38]. Dual-DCN [28] employed dual-path dense convolutional network with the depth of 6 that produced a run-time complexity of $O(2 \cdot 6^2)$. The proposed fusion network included three-path dense convolutional networks with the same depth by fusing back- and front-end differential network architectures, resulting in a run-time complexity of $O(3 \cdot 6^2)$. In the inference stage, a two-steps decision for the proposed made the run-time be $O(2 \cdot (3 \cdot 6^2))$ that gave it an expensive computational complexity while producing a better result.

## 5. Conclusions

This paper presented a robust fusion network for detecting changed/unchanged areas in high-resolution panchromatic images. The proposed method learns and identifies the changed/unchanged areas by combining front- and back-end neural network architectures. The dual outputs are efficiently incorporated for low- and high-level differential features with a modified loss function that combines the contrastive and weighted binary cross entropy losses.

In addition, a post-processing step was applied to enhance the sensitivity and specificity from false changes/unchanged detections based on the neighboring information. We found through qualitative and quantitative evaluations that the proposed algorithm can yield a higher sensitivity and specificity compared to the existing algorithms, even under noisy conditions such as geometric distortions and different viewing angles

For further work, the proposed algorithm can be extended for other modalities such as multi-spectrum images, Pan-sharpening, and SAR data. In addition, the proposed algorithm requires expensive time complexity caused by pixel-wise detection with a two-stage decision. To accelerate run-time complexity, block-wise prediction design would also be a focus of future work.

**Author Contributions:** All authors contributed to the writing of the manuscript. W.W. and D.S. conceived, designed the algorithm, and analyzed the data. W.W. developed it and conducted the experiments and analyzed the data. D.S. supervised the study.

**Funding:** This research was supported by the Ministry of Science and ICT (MSIT), Korea, under the Information Technology Research Center (ITRC) support program (IITP-2018-2016-0-00288) supervised by the Institute for Information & communications Technology Promotion (IITP) and Basic Science Research Program through the National Research Foundation of Korea (NRF) funded by the Ministry of Science, ICT & Future Planning (NRF-2018R1A2B2008238).

**Conflicts of Interest:** The authors declare no conflicts of interest.

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
