# Peer review of "Fusion Network for Change Detection of High-Resolution Panchromatic Imagery"

_applsci, doi:10.3390/app9071441_

Reviewer 1 Report

This paper proposes to combine the results from a late-fusion network and an early fusion network for change detection in geographical images. The writing is generally clear. My concerns are as follows.

Technical parts:

- Are “front-end network” and “back-end network” existing terms to describe the two types of networks? I personally think they are not very clear. “front-end” and “back-end” could be understood as two sequential parts of a single network, but in this paper they are two networks jointly trained and make parallel decisions. It seems to me that “early fusion” and “late fusion” seems to be more precise.

- line 189: “where ?^1 and ?^2 are the probability outputs of ?^1 and ?^2, respectively.”

Are P^1 and P^2 probabilities or features? On the one hand, Eq. 2 seems to be distances between two feature vectors. On the other hand, in the context of change detection, what does the probability for a single image represent? Please clarify.

- line 175 – 176: A dense convolutional connection is employed in the proposed fusion network …

Here I suggest to cite the first paper proposing such densely-connected structures:

G. Huang, et al. Densely connected convolutional networks. In CVPR 2017.

- line 166: Such an architecture uses low-level differential features that are relatively sensitive to noises because they produce true positives and more false positives.

I think better discussion and clarification are needed here.

First, I think it is not very precise to say FDN “uses low-level differential features”. It does concatenate low-level features as input, but the decision is based on deep features at the end of the network. Some other places in the paper also have similar interpretations. It would be good if the authors use a more precise description.

Second, I think the rationale here is not persuasive. For me, more false positives should be the result of features “relatively sensitive to noises”, not the cause. I would like to see more discussions on the reason why FDN is more sensitive to noise; some intuitive understandings are sufficient for me.

- Fig. 1: it would be good to label the name FDN and BDN in the figure alone with their corresponding networks. Although this is described in the paper, it makes the figure more clear with the network name on it.

Experiments:

- The proposed method includes some components: fusing two networks and a two-stage decision process. I would like to see some ablation analysis to validate the effects of these components. This provides better understanding of the proposed network. For example:

(1) what’s the performance of each individual network, compared to fusing the two networks?

(2) What’s performance of the two-stage decision compared to other decision methods, e.g. decision based on max/mean between the output of the two networks?

(3) Can we train a fusion network to fuse the output of the two networks to make decision? What is the performance compared to the proposed two-stage decision process?

Others:

- line 215: The value of ? is set to 0.7 to further penalize ??. is to prevent …

Seems to be a typo

Author Response

Technical parts:

1.          Are “front-end network” and “back-end network” existing terms to describe the two types of networks? I personally think they are not very clear. “front-end” and “back-end” could be understood as two sequential parts of a single network, but in this paper they are two networks jointly trained and make parallel decisions. It seems to me that “early fusion” and “late fusion” seems to be more precise.

Ă°  Thank you for your comment. Yes, “front-end network” and “back-end network” are our term to distinguish early and late comparison between two images. We have described it in the section 2 and drawn in Figure 2.

2.          line 217: “where ?^1 and ?^2 are the probability outputs of ?^1 and ?^2, respectively.”

Are P^1 and P^2 probabilities or features? On the one hand, Eq. 2 seems to be distances between two feature vectors. On the other hand, in the context of change detection, what does the probability for a single image represent? Please clarify.

Ă°      In this paper performs pixel-wise detection by inspecting local neighborhood pixels information. P^1 and P^2 are the probability outputs of each ?^1 and ?^2 branch network which corresponds to the central pixel of each patch image input (I). Eq. 2 is the distance between two probability output indicated change pixel for the central pixel of patch image. It was applied by sliding the window detection.

Ă°  See line (214-220)

3.          Line 203 – 205: A dense convolutional connection is employed in the proposed fusion network. Here I suggest to cite the first paper proposing such densely-connected structures. Huang, et al. Densely connected convolutional networks. In CVPR 2017. 

Ă°  Thank you for your suggestion. I have cited the reference of densely-connected structure.

Ă°  See line 203 – 205 and reference [32]

“A dense convolutional connection is employed in the proposed fusion network to enhance the feature representation [32].”

4.          line 190: Such an architecture uses low-level differential features that are relatively sensitive to noises because they produce true positives and more false positives

First, I think it is not very precise to say FDN “uses low-level differential features”. It does concatenate low-level features as input, but the decision is based on deep features at the end of the network. Some other places in the paper also have similar interpretations. It would be good if the authors use a more precise description

Second, I think the rationale here is not persuasive. For me, more false positives should be the result of features “relatively sensitive to noises”, not the cause. I would like to see more discussions on the reason why FDN is more sensitive to noise; some intuitive understandings are sufficient for me.

Ă°    FDN conducts the decision in the end of network. However, by concatenating of both low-level features of images in the early network, it performs early comparison between two low-level of images. And then, it learns dependent learning together in a single network. The difference of both inputs is extracted since low-level features. It would be very hard to learn for invariant changes. In contrast to BDN, it employs dual network. It is expected to be independently learned for each path network by extracting the difference of high-level features in the end of network.

Ă°    FDN is relative sensitive to noise caused by direct low-level features comparison. As we mentioned that this noise can be caused by misalignment of geometric error and different angle view of two temporal images. In addition, resulting more false positive are caused by dependent learning of both features together in a single network. It leads to hard learning the features producing a global change detection.

However, we have modified more precise description as follow:

See line 189-195

“For a change detection, an FDN architecture is commonly used for identifying changed pixels. Such an architecture uses low-level differential features that are relatively sensitive to noises. It is caused by direct low-level features comparison which misalignments of geometric error and different angle view are very influential. This FDN assigns DI or JF to a single path network. They conduct dependent learning of both low-level features together which lead to hard learning for invariant changes and above-mentioned noisy condition. Thus, this approach would produce a global change detection, resulting true positives and more false positives.”          

5.          Fig. 1: it would be good to label the name FDN and BDN in the figure alone with their corresponding networks. Although this is described in the paper, it makes the figure more clear with the network name on it

Ă°    Thank you for your comments. Figure 1 is a general CNN architecture. However, we have added FDN and BDN figures on Figure 2.

Experiments:

6.          The proposed method includes some components: fusing two networks and a two-stage decision process. I would like to see some ablation analysis to validate the effects of these components. This provides better understanding of the proposed network. For example:

1)       what’s the performance of each individual network, compared to fusing the two networks?

Ă°  Thank you for your suggestion. We have added the experiment result regarding the proposed two-stage decision and each individual network output. See at Table 2 and Figure 10

2)       What’s performance of the two-stage decision compared to other decision methods, e.g. decision based on max/mean between the output of the two networks?

Ă°  The goal of the proposed two stage decision is designed to ensure the prediction of two network outputs by operating intersection and union operation. The Intersection operation is used for strictly detection to avoid false positives. In addition, the union in second prediction is employed to solve the false negative. However, we have compared it to mean operation as you suggested. See at Table 2 and Figure 10.

3)       Can we train a fusion network to fuse the output of the two networks to make decision? What is the performance compared to the proposed two-stage decision process?

Ă°    Thank you for your comments. We have added the evaluation by comparing single output fusion network (SOFN). This network fused the output of the two networks to a single output. We trained this network with the same training parameters and dataset. Objective and subjective evaluation are presented in Table 2 and Figure 10, respectively.

Others:

7.          Line 243: The value of ? is set to 0.7 to further penalize ??. is to prevent …

Seems to be a typo

Ă°    Thank you for your correction. We have revised it.

“The value of ? is set to 0.7 to further penalize ??. It is to prevent …”

Reviewer 2 Report

Overall, the present paper represents a interesting contribution to CD based on Deep Learning and Data Fusion. However, before publication, it is my opinion that the following major comments should be addressed by the authors:

1)      The abstract should be rephrased aiming at improved clarity.

2)      Sec. I – Please explain in more detail the sentence “Local change vectors have also been used by applying neighbor pixels to avoid a direct subtraction based on the log ratio”

3)      The following related works on hypothesis testing approaches for CD should be discussed for the sake of a complete review of the literature:

[R1]  "On multiple covariance equality testing with application to SAR change detection." IEEE Transactions on Signal Processing 65.19 (2017): 5078-5091

[R2] D. E. Wahl et al. "A new maximum-likelihood change estimator for two-pass SAR coherent change detection." IEEE Transactions on Geoscience and Remote Sensing 54.4 (2016): 2460-2469.

[R3] "DECHADE: DEtecting slight Changes with HArd DEcisions in Wireless Sensor Networks." International Journal of General Systems 47.5 (2018): 535-548.

4)      Please add a notation paragraph at the end of Sec. I.

5)      The description of the contributions should be rephrased so as to improve its effectiveness, e.g. possibly with a bullet list.

6)       On the statement “The CNN architecture employs multiple convolutional layers, followed by rectified linear units (ReLUs), resulting in the development of feature maps.” -> CNN architectures do not necessarily have ReLUs as activation functions. Please rectify this sentence.

7)       Please discuss the computational complexity of the proposed fusion architecture.

8)       By looking at Sec. 3.3 (and, specifically, Fig.3 ), it seems that the authors are using a AND decision fusion rule, which is a specific case of the more general counting rule, e.g

[R4] Viswanathan, R., and Valentine Aalo. "On counting rules in distributed detection." IEEE Transactions on Acoustics, Speech, and Signal Processing 37.5 (1989): 772-775.

[R5] "A systematic framework for composite hypothesis testing of independent Bernoulli trials." IEEE Signal Processing Letters 22.9 (2015): 1249-1253.

The above works could mentioned for completeness. Additionally, it is not completely clear to me how the flowcharts in Fig. 3 and Fig. 4 are combined.

9)       Fig. 7 is very hard to read. Please improve its readability.

10)  Conclusions should be enriched with further avenues of research.

Author Response

1.          The abstract should be rephrased aiming at improved clarity.

Ă°  Thank you for your comment. The abstract has been reprased  to improve clarity.

“This paper proposes a fusion network for detecting changes between two high-resolution panchromatic images. The proposed fusion network consists of front- and back-end neural network architectures to generate dual outputs for change detection. Two networks for change detection are applied to handle image- and high-level changes of information, respectively. The fusion network employs single-path and dual-path networks to accomplish low-level and high-level differential detection, respectively. Based on two dual outputs, a two-stage decision algorithm is proposed to efficiently yield the final change detection results. The dual outputs are incorporated to the two-stage decision by operating logical operations. The proposed algorithm was designed to incorporate not only dual network outputs but also neighboring information. In this paper, a new fused loss function is presented to estimate the errors and optimize the proposed network during the learning stage. Based on our experimental evaluation, the proposed method yields a better detection performance than conventional neural network algorithms, with an average area under the curve of 0.9709, percentage correct classification of 99%, and Kappa of 75 for many test datasets.”

2.          Sec. I – Please explain in more detail the sentence “Local change vectors have also been used by applying neighbor pixels to avoid a direct subtraction based on the log ratio”

Ă°    Thank you for your comment. We have modified it by describing more detail.

“Local change vectors have also been used by applying neighbor pixels to avoid a direct subtraction based on the log ratio. This method computes a mean value of the log ratio of temporal neighbor pixels.”

3.          The following related works on hypothesis testing approaches for CD should be discussed for the sake of a complete review of the literature:

a)       R1]  "On multiple covariance equality testing with application to SAR change detection." IEEE Transactions on Signal Processing 65.19 (2017): 5078-5091

b)      R2] D. E. Wahl et al. "A new maximum-likelihood change estimator for two-pass SAR coherent change detection." IEEE Transactions on Geoscience and Remote Sensing 54.4 (2016): 2460-2469.

c)       [R3] "DECHADE: DEtecting slight Changes with HArd DEcisions in Wireless Sensor Networks." International Journal of General Systems 47.5 (2018): 535-548.

Ă°  Thank you for your comments. The above references are the papers about change detection with statistic approach. Moreover, the R3 is change detection for wireless sensor. However, our paper focus on solving change detection problem on remote sensing imagery by deep learning approach. All existing works described in the paper should be related to change detection with deep learning approaches for remote sensing image.

4.          Please add a notation paragraph at the end of Sec. I.

Ă°  “We tried to find the notation paragraph from the guideline of MDPI manuscript. However, the guideline does not ask us to write the notation paragraph. However, we mentioned all the notations in the proper positions that the notations are used in the manuscript. If you think the notation paragraph is really needed in the revised manuscript, please show some examples for notation paragraphes.”

5.          The description of the contributions should be rephrased so as to improve its effectiveness, e.g. possibly with a bullet list.

Ă°  Thank you for your comment. We have added the paper contribution in the end of section 1.

“This work contributes three main key features as follows. (1) Unlike the mentioned existing works above, we propose a fusion network by combining a front- and back-end networks to perform the low- and high-level differential detection in one structure. (2) A combining loss function between contrastive loss and binary cross entropy loss is proposed to accomplish fusion of the proposed networks in training stage. (3) The two-stage decision as a post-processing is presented to validate and ensure the changes prediction at the inference stage to obtain better the final change map.”

6.          On the statement “The CNN architecture employs multiple convolutional layers, followed by rectified linear units (ReLUs), resulting in the development of feature maps.” -> CNN architectures do not necessarily have ReLUs as activation functions. Please rectify this sentence.

Ă°  Thank you for your comment. We have rectified the sentence.

“The CNN architecture employs multiple convolutional layers, followed by an activation function, resulting in the development of feature maps. Rectified linear unit (ReLU) is widely used as the activation function in many CNN architectures. To progressively gather global spatial information, the feature maps are sub-sampled by the pooling layer. The final feature maps are connected to a fully connected layer to produce the class probability outputs 

7.          Please discuss the computational complexity of the proposed fusion architecture.

Ă°  Thank you for your comment. We have added a discussion paragraph regarding time complexity of the proposed network.

“Regarding time complexity, the proposed fusion network consumes more computational complexity than the existing algorithm by a factor of approximately two over the dual-path network and three with single-path network. It is due to the proposed architecture designed with more network paths. In addition, the proposed two-stage decision requires additional prediction process in the inference stage.”

8.          By looking at Sec. 3.3 (and, specifically, Fig.4 ), it seems that the authors are using a AND decision fusion rule, which is a specific case of the more general counting rule, e.g.

a.    R4] Viswanathan, R., and Valentine Aalo. "On counting rules in distributed detection." IEEE Transactions on Acoustics, Speech, and Signal Processing 37.5 (1989): 772-775.

b.    R5] "A systematic framework for composite hypothesis testing of independent Bernoulli trials." IEEE Signal Processing Letters 22.9 (2015): 1249-1253.

The above works could mentioned for completeness. Additionally, it is not completely clear to me how the flowcharts in Fig. 4 and Fig. 5 are combined.

Ă°  Thank you for your comments. This paper proposed two-stage decision for change detection. The proposed fusion network produces dual output, namely D and O. Each output will generate the decision for change detection. Based on those decision, a logical operation is operated in inference stage. This is a sequential process which flowchart of figure 5 is performed after flowchart of figure 4 is finished. Firstly, intersection/AND operation is applied in flowchart of figure 4 to get strictly prediction. It is conducted in the entire image by pixel-wise sliding in the raster scan order. Then, the second prediction of flowchart of figure 5 is applied by operating union/OR operation to validate and ensure true positive detection with the same way of sliding in the raster scan order detection. Finally, the final result is obtained based on result of figure 5 flowchart.  

9.          Fig. 8 is very hard to read. Please improve its readability.

Ă°  Thank you for your comment. We have changed the figure 8.

10.      Conclusions should be enriched with further avenues of research.

Ă°  Thank you for your comment. We have added further avenue of research.

“This paper presented a robust fusion network for detecting changed/unchanged areas in high-resolution panchromatic images. The proposed method learns and identifies the changed/unchanged areas by combining front- and back-end neural network architectures. The dual outputs are efficiently incorporated for low- and high-level differential features with a modified loss function that combines the contrastive and weighted binary cross entropy losses. In addition, a post-processing step is applied to enhance the sensitivity and specificity from false changes/unchanged detections based on the neighboring information. We found through qualitative and quantitative evaluations that the proposed algorithm can yield a higher sensitivity and specificity compared to the existing algorithms, even under noisy conditions such as geometric distortions and different viewing angles. In addition, time acceleration by designing block-wise prediction would be focus as the future work.”

Reviewer 3 Report

In this paper, the author has proposed a fusion network algorithm (combining front- and back-end networks) to detect the changed/unchanged areas in high-resolution panchromatic images. 

 Each network has been allocated to a specific task (one for handling images and the other one for high-level changes of information).

The algorithm and results sound logical, and the comparison shows how the proposed algorithm outperforms other methods.

- Quality of the figures should be improved (figures 2 and 7). 

Author Response

1.         Quality of the figures should be improved (figures 3 and 8).

Ă°  Thank you for your comment. We have improved quality of the Figure. 3 and 8

Round  2

Reviewer 1 Report

Most of my concerns have been addressed. I have a single question here:

Previous Comments and answers:

Are P^1 and P^2 probabilities or features? On the one hand, Eq. 2 seems to be distances between two feature vectors. On the other hand, in the context of change detection, what does the probability for a single image represent? Please clarify.

Ă°      In this paper performs pixel-wise detection by inspecting local neighborhood pixels information. P^1 and P^2 are the probability outputs of each ?^1 and ?^2 branch network which corresponds to the central pixel of each patch image input (I). Eq. 2 is the distance between two probability output indicated change pixel for the central pixel of patch image. It was applied by sliding the window detection.

My question:

This is still not very clear for me. If P^1 and P^2 are probability maps, they are probability of what? The probability of changing happens at a pixel? But the features from the two streams have not been compared at the stage of P1 and P2. How do you determine the probability of change at the end of each individual stream?

Author Response

Previous Comments and answers:

Are P^1 and P^2 probabilities or features? On the one hand, Eq. 2 seems to be distances between two feature vectors. On the other hand, in the context of change detection, what does the probability for a single image represent? Please clarify.

Ă°      In this paper performs pixel-wise detection by inspecting local neighborhood pixels information. P^1 and P^2 are the probability outputs of each ?^1 and ?^2 branch network which corresponds to the central pixel of each patch image input (I). Eq. 2 is the distance between two probability output indicated change pixel for the central pixel of patch image. It was applied by sliding the window detection.

My question:

This is still not very clear for me. If P^1 and P^2 are probability maps, they are probability of what? The probability of changing happens at a pixel? But the features from the two streams have not been compared at the stage of P1 and P2. How do you determine the probability of change at the end of each individual stream?

Ă°      Thank you for your comment. Many thanks to correct us, the P^1 and P^2 are the outputs of each individual stream activated by sigmoid function. A change at pixel is determined by calculating the dissimilarity distance (D) between those outputs P^1 and P2 using Euclidean distance. If the value of D is close to 1 then a pixel is changed. Therefore, we have corrected our description (see line 229)

Reviewer 2 Report

Overall, the present paper represents a interesting contribution to CD based on Deep Learning and Data Fusion. Additionally, the authors have somewhat improved their manuscript since its first submission.

However, in some relevant cases, they have strived for the minimum effort when replying to the listed comments and revising the paper accordingly. For the mentioned reason, I recommend another major revision round so that they can cope with the following remaining comments in an appropriate fashion:

1)      The following related works on hypothesis testing approaches for CD are needed to be discussed for the sake of a complete review of the literature. Indeed, please notice that, when introducing a novel approach for CD, not only the works tackling CD with the same technique (e.g. Deep Learning) should be discussed, but also other approaches tackling the same problem with different philosophies.

[R1]  "On multiple covariance equality testing with application to SAR change detection." IEEE Transactions on Signal Processing 65.19 (2017): 5078-5091

[R2] D. E. Wahl et al. "A new maximum-likelihood change estimator for two-pass SAR coherent change detection." IEEE Transactions on Geoscience and Remote Sensing 54.4 (2016): 2460-2469.

[R2] "DECHADE: DEtecting slight Changes with HArd DEcisions in Wireless Sensor Networks." International Journal of General Systems 47.5 (2018): 535-548.

2)      In my opinion, a notation paragraph is quite helpful to the generic reader. Indeed, this collects all the mathematical notations and symbols used/employed throughout the paper. Though not required in general, it is a common practice to add it when a number of equations are present in a certain work.

3)       The provided discussion on the computational complexity of the proposed fusion architecture, although valuable, it is still not satisfactory. The authors should better detail it in terms of the relevant parameters and layers used, by resorting to the well-known “big O” formulation.

4)       Since this is a sequential process, why did the authors not provide Figs. 4 and 5 into a single unified workflow?  Additionally, I could not find any mention to the following works, tackling decision fusion with the well-known counting rule, which comprises AND and OR strategies as “special cases”, and which may be helpful to support the considered per-pixel strategies.

[R3] Viswanathan, R., and Valentine Aalo. "On counting rules in distributed detection." IEEE Transactions on Acoustics, Speech, and Signal Processing 37.5 (1989): 772-775.

[R4] "A systematic framework for composite hypothesis testing of independent Bernoulli trials." IEEE Signal Processing Letters 22.9 (2015): 1249-1253.

5)       Fig. 8 is still very hard to read. Probably, the use of vectorized images (.eps) would greatly improve their readability.

6)      The description of further avenues of research in “Conclusions” section is still unsatisfactory and should be significantly improved.

Author Response

Responses to Reviewers’ Comments

Title: Fusion Network for Change Detection of High-Resolution Panchromatic Imagery

First of all, we appreciate your thoughtful comments on the paper listed bellowed. For the revised version, we admit that there were not correctly sentences and statement. Thus, we revised and deleted some ambiguous statements based on reviewers’ comments. In addition, we added additional evaluation and analysis regarding computational complexity. For better understanding, we have replaced several figures by new ones for better quality.

#Round 2

Overall, the present paper represents a interesting contribution to CD based on Deep Learning and Data Fusion. Additionally, the authors have somewhat improved their manuscript since its first submission.

However, in some relevant cases, they have strived for the minimum effort when replying to the listed comments and revising the paper accordingly. For the mentioned reason, I recommend another major revision round so that they can cope with the following remaining comments in an appropriate fashion:

1)      The following related works on hypothesis testing approaches for CD are needed to be discussed for the sake of a complete review of the literature. Indeed, please notice that, when introducing a novel approach for CD, not only the works tackling CD with the same technique (e.g. Deep Learning) should be discussed, but also other approachlies tackling the same problem with different philosophies

[R1]  "On multiple covariance equality testing with application to SAR change detection." IEEE Transactions on Signal Processing 65.19 (2017): 5078-5091

[R2] D. E. Wahl et al. "A new maximum-likelihood change estimator for two-pass SAR coherent change detection." IEEE Transactions on Geoscience and Remote Sensing 54.4 (2016): 2460-2469

[R3] "DECHADE: DEtecting slight Changes with HArd DEcisions in Wireless Sensor Networks." International Journal of General Systems 47.5 (2018): 535-548

Ă°  Thank you for your comment. We have mentioned [R1], [R2], and [R3] in Section 1.

Ă°  See lines 31-40

2)      In my opinion, a notation paragraph is quite helpful to the generic reader. Indeed, this collects all the mathematical notations and symbols used/employed throughout the paper. Though not required in general, it is a common practice to add it when a number of equations are present in a certain work.

Ă°  Thank you for your suggestion. We added Table 1 to describe all notations.

3)       The provided discussion on the computational complexity of the proposed fusion architecture, although valuable, it is still not satisfactory. The authors should better detail it in terms of the relevant parameters and layers used, by resorting to the well-known “big O” formulation.

Ă°  Thank you for your suggestion. We added analysis of time complexity in term of big O formulation in the end of Section 4 (See lines 367-378). We analyzed the generic formulation of big O by citing the existing paper [38] regarding complexity analysis of DenseNet. We cannot provide detail big O analysis for each process and parameters because we want to evaluate the generic process of the proposed algorithm. Please understand that it is not easy to formulate entire process of complex deep CNN architecture. We believe that by evaluating in general formulation, readers can understand and compare how expensive the time complexity of the proposed algorithm is.

4)       Since this is a sequential process, why did the authors not provide Figs. 4 and 5 into a single unified workflow?  Additionally, I could not find any mention to the following works, tackling decision fusion with the well-known counting rule, which comprises AND and OR strategies as “special cases”, and which may be helpful to support the considered per-pixel strategies.

[R3] Viswanathan, R., and Valentine Aalo. "On counting rules in distributed detection." IEEE Transactions on Acoustics, Speech, and Signal Processing 37.5 (1989): 772-775

[R4] "A systematic framework for composite hypothesis testing of independent Bernoulli trials." IEEE Signal Processing Letters 22.9 (2015): 1249-1253.

Ă°  Thank you for your comment. This paper proposed two-stage decision in the inference stage. There are two main steps to identify a change map including first and second predictions. The first prediction is applied in the entire image with sliding in the raster scan order, as shown in Fig. 4. After finish, the sliding scan is repeated with the second prediction, as shown in Fig. 5. We separated it to be Figs. 4 and 5 because the proposed contains two main steps. We thought that it could be easier to understand the scan order position (x,y).

Ă°  We have mentioned reference [R3] (See lines 261-263). Regarding the decision fusion, we used the hard decision rule which means by the same threshold of both outputs (D and O).

5)       Fig. 8 is still very hard to read. Probably, the use of vectorized images (.eps) would greatly improve their readability.

Ă°  Thank you for your comment. We enhanced all figures with the vectorized image format.

6)      The description of further avenues of research in “Conclusions” section is still unsatisfactory and should be significantly improved.

Ă°  Thank you for your comment. We have enhanced the further avenues research in conclusion section (See lines 390-393 ).

Again, we would like to offer our sincere thanks to the three reviewers for their efforts and making valuable input and suggestions for improving the manuscript. We hope the revised manuscript will be published based on your comments.
